# Xuebijing Administration Alleviates Pulmonary Endothelial Inflammation and Coagulation Dysregulation in the Early Phase of Sepsis in Rats

**DOI:** 10.3390/jcm11226696

**Published:** 2022-11-11

**Authors:** Jie Lv, Xiaoxia Guo, Huiying Zhao, Gang Zhou, Youzhong An

**Affiliations:** Department of Critical Care Medicine, Peking University People’s Hospital, Beijing 100033, China

**Keywords:** Xuebijing, sepsis, endothelium, cytokines, disseminated intravascular coagulation

## Abstract

Ethnopharmacological relevance: Xuebijing injection is a Chinese herbal-derived drug composed of radix paeoniaerubra, rhizomachuanxiong, Salvia miltiorrhiza, floscarthami, and Angelica sinensis. This study aimed to investigate the effects of Xuebijing administration on pulmonary endothelial injury and coagulation dysfunction in a cecal ligation and puncture (CLP)-induced sepsis rat model. Materials and methods: A CLP-induced sepsis rat model was established. The CLP rats were treated with a vehicle or Xuebijing via intravenous infusion and sacrificed at 2, 4, 6, 8, or 12 h after CLP for lung tissue and blood sample collection. The mean arterial pressure (MAP) was monitored. Transmission microscopy examination and H&E staining were performed to observe pulmonary structural alterations. Enzyme linked immunosorbent assay (ELISA) was performed to measure the plasma levels of epithelial markers, proinflammatory cytokines, and coagulation-related proteins. Results: Compared with vehicle treatment, Xuebijing administration maintained the MAP in the normal range until 11 h after CLP. Transmission microscopy and H&E staining revealed that Xuebijing administration alleviated alveolar–capillary barrier impairments and lung inflammation in CLP rats. ELISA showed that Xuebijing administration effectively reversed CLP-induced elevations in the plasma levels of epithelial markers endothelin-1 and von Willebrand factor, starting 6 and 8 h after CLP, respectively. Xuebijing administration also significantly abolished CLP-induced rises in circulating proinflammatory cytokines interleukin 6 (IL-6) at 6 h after CLP, IL-1β at 2 and 12 h after CLP, and TNF-α at 2, 4, 6, 8, and 12 h after CLP. In addition, Xuebijing administration strongly reversed CLP-induced alterations in circulating active protein C and tissue-type plasminogen activator, starting 4 h and 2 h after CLP, respectively. Conclusions: Xuebijing ameliorates pulmonary endothelial injury, systemic inflammation, and coagulation dysfunction in early sepsis.

## 1. Introduction

Sepsis is a life-threatening multiple organ dysfunction caused by the inflammatory response to infection, affecting approximately 31.5 million individuals and contributing to 19.4 million deaths worldwide each year [1]. The activation of coagulation commonly occurs in sepsis as a host immune response to infection and can progress to disseminated intravascular coagulation (DIC). DIC is a widespread hypercoagulable state characterized by vascular clotting, rapidly leading to multiple organ dysfunction syndrome and death if early diagnosis and treatment fail to occur [2]. Nearly 30% to 50% of cases of severe sepsis develop DIC, and the mortality rate doubles in septic patients if they also suffer from DIC [3]. Therefore, early prevention, diagnosis, and treatment for DIC in septic patients are critical for their survival.

The pathophysiology of sepsis-induced coagulopathy and DIC involves coagulation activation, fibrinolysis suppression, endothelial dysfunction, platelet aggregation, and the impairment of anticoagulant systems [4]. During sepsis, the pathogen-associated molecular patterns trigger systemic inflammatory responses, leading to an excess production of proinflammatory cytokines that are responsible for coagulopathy, such as interleukin (IL)-1, IL-6, and tumor necrosis factor-α (TNF-α) [5]. The vascular endothelium is the main target of microorganisms and inflammatory stimuli in sepsis, and endothelial injury represents one of the unique features of sepsis-induced DIC [4]. Vascular endothelial cells modulate coagulation and fibrinolysis by secreting multiple factors, such as endothelin-1 (ET-1), von Willebrand factor (VWF), protein C, and tissue-type plasminogen activator (t-PA). These factors are considered as important biomarkers for epithelial dysfunction and coagulopathy in sepsis [6].

Xuebijing injection is a Chinese herbal-derived drug composed of radix paeoniaerubra, rhizomachuanxiong, Salvia miltiorrhiza, floscarthami, and Angelica sinensis, containing active ingredients, including paeoniflorin, ferulic acid, danshensu, and hydroxysafflor yellow [7]. Owing to its anti-inflammatory property [8,9,10,11], Xuebijing has been used for the treatment of sepsis-induced serious lung infections, systemic inflammatory response syndrome, and multiple organ dysfunction syndrome in China [12,13,14]. In 2004, Xuebijing injection was approved by the China Food and Drug Administration to treat sepsis (#Z20040033). Clinical study has shown that Xuebijing injection can improve coagulopathy and treat DIC in patients with severe sepsis [15,16]. Xuebijing has also been shown to prevent organ injuries and improve survival in heatstroke mice by attenuating inflammatory responses and endothelial injury [17]. However, little is known about whether and how Xuebijing ameliorates endothelial injury and coagulopathy in the early phase of sepsis.

In this study, using a cecal ligation and puncture (CLP)-induced sepsis rat model, we investigated the protective effects of Xuebijing against the mean arterial pressure (MAP) declines, pulmonary endothelial injury, inflammation, and coagulation dysfunction in early sepsis. Our results suggest that Xuebijing is a promising therapeutic agent for preventing the development of endothelial injury and coagulopathy in early sepsis.

## 2. Materials and Methods

### 2.1. Animals

This study was approved by the Research Ethics Board at Peking University People’s Hospital (#2018PHC003; Beijing, China). All animal experiments were conducted according to the Guidelines of Animal Research of Peking University People’s Hospital. A total of 75 specific pathogen-free Sprague–Dawley rats weighing 300–400 g were purchased from the Experimental Animal Center of Peking University. After a week of acclimatization, the rats were randomly divided into sham, CLP, and CLP + Xuebijing groups (n = 25/group).

Sham or CLP operation was performed as previously described [18]. Briefly, the rats were anesthetized with 10% chloral hydrate (4 mL/kg; Tianjin Chase Sun Pharmaceutical Co., Ltd., Tianjin, China). The right internal carotid artery and the right femoral vein were exposed. The MAP was monitored through the right internal carotid artery using an electrocardiogram monitor (iPM8; Mindray, Shenzhen, China). A catheter was inserted into the right femoral vein for drug administration. An incision was made in the middle of the abdomen to expose the cecum. Then, the cecum was ligated and punctured three times using a 16-gauge needle. After returning the cecum into the abdominal cavity, the incision was closed, followed by normal saline infusion at 4 mL/kg/h as maintenance fluids. The sham rats underwent the same procedures except for the ligation and punctures. The CLP + Xuebijing group was administered 4 mL/kg Xuebijing (Tianjin Chase Sun Pharmaceutical Co., Ltd.) for 1 h through the right femoral vein via intravenous infusion after closing the abdominal cavity, followed by normal saline infusion at 4 mL/kg/has maintenance fluids. The CLP group was administered normal saline as a vehicle treatment.

### 2.2. Sample Collection

Rats in each group were randomly divided into 2, 4, 6, 8, or 12 h subgroups (n = 5/subgroup), and blood samples were collected from the carotid artery at each time point after the operation. Plasma samples were obtained by centrifuging the blood samples at 3000 r/min for 5 min and stored at −80 ℃. Rats were then sacrificed. Lung tissue samples of around 0.5 × 0.5 × 0.3 cm^3^ were collected and stored in glutaraldehyde.

### 2.3. Transmission Electron Microscopy

The lung tissue samples were fixed with 2.5% glutaraldehyde in 0.01 mol/L phosphate buffer at 4 °C, followed by incubation with 2% osmium tetroxide. The sample was then dehydrated in graded ethanol solutions, embedded in paraffin, sliced, and stained with 3% uranyl acetate and lead citrate. The slides were observed using a transmission electron microscope (Tecnai Spirit Bio TWIN; Zeiss, Oberkochen, Germany) at 80 kV.

### 2.4. Hematoxylin and Eosin (H&E) Staining

The lung tissue samples were fixed with embedding fixative for more than 24 h, embedded in paraffin, cut into 4 μm thick sections, and subjected to H&E staining. The endothelial structural changes and inflammation were examined.

### 2.5. Enzyme-linked Immunosorbent Assay (ELISA)

Circulating levels of ET-1 (EH0205; Zsbabio, Beijing, China), VWF (ER322; Rapidbio), IL-1β (88-7261; Invitrogen, Carlsbad, CA, USA), TNF-α (88-7340; Invitrogen), active protein C (APC; EH460; Rapidbio), and t-PA (EH411; Rapidbio) were measured using corresponding ELISA kits according to the manufacturers’ instructions.

### 2.6. Statistical Analysis

Data are presented as the mean ± standard deviation. Statistical analysis was conducted using SPSS 20.0 software (IBM, Armonk, NY, USA). Differences among groups were compared using the one-way analysis of variance, followed by *t*-test. A *p* value less than 0.05 was considered as statistically significant.

## 3. Results

### 3.1. Xuebijing Administration Restores MAP in CLP Rats

To explore the effect of Xuebijing administration on the hemodynamics in early septic shock, we recorded the MAP in three randomly selected rats from the 12 h subgroup in each group. As shown in Supplemental Appendix A and Figure 1, the MAP in the sham group remained in the normal range (>70 mmHg) within 12 h after operation, whereas the MAP in the CLP group gradually decreased below 70 mmHg at 9 h after operation. The 11 h and 12 h MAP values in the CLP group were significantly lower than those in the sham group (both *p* < 0.01). These data suggest that CLP induces a septic shock in the CLP group. Although the MAP in the CLP + Xuebijing group exhibited a similar descending trend to that in the CLP group, the MAP in the CLP + Xuebijing group remained in the normal range (>70 mmHg) until 10 h after operation. In addition, the 12 h MAP values in the CLP + Xuebijing group were significantly greater than those in the CLP group (*p* < 0.05). This finding suggests that, compared with vehicle treatment, Xuebijing administration can restore MAP in early septic shock in CLP rats.

### 3.2. Xuebijing Alleviates Pulmonary Endothelial Impairments and Inflammation in CLP Rats

Sepsis-induced endothelial dysfunction is considered to be the key factor in the progression from sepsis to organ failure [19]. Thus, we employed transmission microscopy and H&E staining to examine the structure of the pulmonary endothelium in rats. Transmission microscopy examination showed that the sham rats had an intact alveolar–capillary barrier composed of the capillary basement membrane, connective tissue, and pulmonary basement membrane (Figure 2A(a)), whereas the CLP rats exhibited a variety of alterations in the alveolar–capillary barrier, including pulmonary edema, endosomes in type I alveolar epithelial cells, and an impaired alveolar–capillary barrier (Figure 2A(b,c)). Alveolar–capillary barrier disruption was also present in Xuebijing-treated rats; however, the number of mitochondria was increased in the alveolar epithelial cells of Xuebijing-treated rats compared with those of vehicle-treated rats (Figure 2A(d)). This finding suggests that Xuebijing promotes mitochondria biogenesis in septic rats.

In addition, H&E staining revealed that, compared with the lung tissue from the sham rats (Figure 2B(a)), the lung tissue from vehicle-treated CLP rats exhibited significant alterations, including alveolar septal thickening, capillary congestion, and inflammatory cell infiltration (Figure 2B(b)). Compared with vehicle treatment, Xuebijing treatment resulted in reductions in capillary congestion and inflammatory cell infiltration (Figure 2B(c)). Taken together, these results suggest that Xuebijing ameliorates sepsis-induced lung injury in CLP rats.

### 3.3. Xuebijing Administration Reduces Circulating Endothelial Markers in CLP Rats

To further investigate the effect of Xuebijing on endothelial dysfunction in CLP rats, we determined the plasma levels of classical endothelial markers ET-1 and VWF [20] in rats. As shown in Appendix A, Figure 2C,D, compared with those in sham rats, ET-1 and VWF plasma levels were increased in vehicle-treated CLP rats in a time-dependent manner. Xuebijing administration partially but significantly reversed CLP-induced elevations in ET-1 and VWF plasma levels, starting 8 and 12 h after administration, respectively (all *p* < 0.05). These data suggest that Xuebijing administration alleviates endothelial cell injury in early sepsis in CLP rats.

### 3.4. Xuebijing Administration Reduces Circulating Proinflammatory Cytokine Levels in CLP Rats

Since sepsis induces systemic inflammation [21], we further explored the effect of Xuebijing administration on the plasma levels of proinflammatory cytokines in rats. As shown in Supplementary Appendix A, Figure 3A,B, compared with those in sham rats, IL-6 and IL-1β plasma levels were significantly increased in vehicle-treated CLP rats starting 6 h after CLP (all *p* < 0.05). TNF-α plasma levels were also significantly elevated in vehicle-treated CLP rats compared with those in sham rats, starting 2 h and peaking at 4 h after CLP (Figure 3C; all *p* < 0.05). These results suggest that CLP induces systemic inflammation in rats. Xuebijing administration significantly abolished CLP-induced elevations in IL-6 (6 h after CLP), IL-1β (2 and 12 h after CLP), and TNF-α (2, 4, 6, 8, and 12 h after CLP) plasma levels at different time points after CLP (all *p* < 0.05), suggesting that Xuebijing administration alleviates sepsis-induced systemic inflammation in CLP rats.

### 3.5. Xuebijing Administration Reverses CLP-Induced Alterations in Circulating APC and t-PA Levels in Rats

The endothelium regulates coagulation and fibrinolysis to maintain blood flow and avoid systemic bleeding or clotting [22]. To explore the protective effect of Xuebijing against sepsis-induced coagulopathy, we determined circulating APC and t-PA levels in rats. As shown in Appendix A, Figure 4A,B, APC plasma levels were significantly decreased in vehicle-treated CLP rats compared with those in sham rats in a time-dependent manner (all *p* < 0.05 except for 2 h after CLP). In contrast, t-PA plasma levels were remarkably increased in vehicle-treated CLP rats compared with those in sham rats, peaking at 2 h after CLP (all *p* < 0.05 except for 8 h after CLP). These results suggest that CLP-induced sepsis drives hemostasis toward a prothrombotic and antifibrinolytic state. Compared with vehicle treatment, Xuebijing administration strongly reversed CLP-induced reductions in APC plasma levels, starting 4 h and peaking at 6 h after CLP (all *p* < 0.05). In addition, Xuebijing administration also potently blunted CLP-induced rises in t-PA plasma levels at 2 and 4 h after CLP (both *p* < 0.05). These results suggest that Xuebijing administration in early sepsis alleviates the sepsis-induced dysregulation of coagulation and fibrinolysis.

## 4. Discussion

In this study, we demonstrated that, in CLP-induced septic rats, Xuebijing administration maintained the MAP and alleviated pulmonary endothelial injury and inflammation, as well as abrogated alterations in the circulating levels of endothelial markers, inflammatory cytokines, and coagulation-related proteins. Our results suggest that Xuebijing may prevent the development of endothelial injury and coagulopathy in the early phase of sepsis.

Septic shock is the most severe complication of sepsis, characterized by persistent arterial hypotension despite the provision of adequate fluid resuscitation [23]. Endothelial barrier dysfunction occurs early in sepsis and leads to the leakage of intravascular proteins and plasma fluids across the compromised endothelial barrier, contributing to poor tissue perfusion and ultimately septic shock [24]. Candidate treatments that can restore and maintain endothelial barrier function in sepsis and septic shock have entered clinical trials [25,26]. In this study, we found that, compared with that in the sham group, the MAP in the CLP group gradually decreased below 70 mmHg starting 9 h after operation despite continuous saline infusion, suggesting that the CLP rats developed septic shock. Although the MAP in the CLP + Xuebijing group exhibited a similar descending trend to that in the CLP group, the MAP in the CLP + Xuebijing group remained in the normal range (>70 mmHg) until 10 h after operation and was significantly higher than that in the CLP group at 12 h after operation. Similarly, Xuebijing has been found to restore the MAP to normal values at 3 h after treatment in a lipopolysaccharide (LPS)-induced septic shock dog model [27]. Together, these findings suggest that Xuebijing administration can maintain the MAP in the normal range in early septic shock, possibly by ameliorating the leakage of vascular fluid, owing to its ability to improve endothelial integrity in sepsis.

In the early phase of sepsis, the pathogen-associated molecular patterns not only destruct the integrity and function of the endothelium, but also stimulate the release of proinflammatory cytokines, vasoconstrictors, and coagulation factors [19,28]. Studies have shown an impaired endothelium, increased micro-vascular permeability, and inflammation in multiple organs in experimental sepsis induced by LPS or CLP, including the kidney, the liver, and the lung [29,30,31]. Researchers have also consistently observed elevated circulating levels of proinflammatory cytokines such as IL-1, IL-6, and TNF-α, as well as endothelial injury markers such as ET-1 and VWF, in septic animal models and patients [32,33,34]. It has been reported that pretreatment with Xuebijing reduces the serum levels of TNF-α, IL-6, and VWF in a dose-dependent manner and attenuates heat-stress-induced endothelial hyper-permeability in heatstroke mice [17,35]. Xuebijing blocks proinflammatory gene expression in lung tissue, reduces pulmonary permeability, and alleviates lung inflammation in septic rats [36]. Consistent with these reports, we demonstrated that Xuebijing effectively alleviated structural impairments and inflammation in the alveolar–capillary barrier and reduced circulating ET-1, VWF, IL-1β, IL-6, and TNF-α in CLP rats within 12 h after sepsis. Sepsis reduces the number of mitochondria and impairs the structure and function of mitochondria, which will further aggravate sepsis [37]. In the present study, we observed increased numbers of mitochondria in the alveolar epithelial cells of Xuebijing-treated rats compared with those of vehicle-treated rats, suggesting that Xuebijing might enhance mitochondrial biogenesis to prevent endothelial injury.

The endothelium plays a vital role in maintaining the balance of coagulation and fibrinolysis in normal hemostasis by synthesizing and expressing pro and anticoagulant molecules [19]. Coagulation activation and fibrinolysis shutdown occur early in patients with septic shock and are associated with an increased mortality. These patients show significantly decreased anticoagulatory APC and increased procoagulatory t-PA serum levels immediately after sepsis onset compared with the healthy controls [38]. Hou et al. have reported that Xuebijing injection significantly elevates platelet counts while reducing the activated partial thromboplastin time, prothrombin time, and thrombin time in septic patients [16]. Yin et al. have shown that Xuebijing injection significantly reduces DIC incidence and 28-day mortality while elevating platelet counts and prothrombin time in severe septic patients [15]. These findings suggest that Xuebijing targets coagulopathy in sepsis; however, the molecular mechanisms remain unclear. In this study, we found that Xuebijing administration potently abrogated sepsis-induced alterations in APC and t-PA secretion in CLP rats in early sepsis, suggesting that Xuebijing modulates the expression and secretion of pro and anticoagulatory factors, possibly by restoring endothelial integrity during sepsis.

## 5. Conclusions

In conclusion, we demonstrated that Xuebijing administration restored the MAP, ameliorated pulmonary endothelial injury and inflammation, suppressed proinflammatory cytokine production, and modulated pro and anticoagulatory factor secretion in the early phase of sepsis in a CLP rat model. These findings suggest that Xuebijing is a promising therapeutic agent in protecting the endothelium and preventing DIC development in early sepsis.

## Figures and Tables

**Figure 1 jcm-11-06696-f001:**
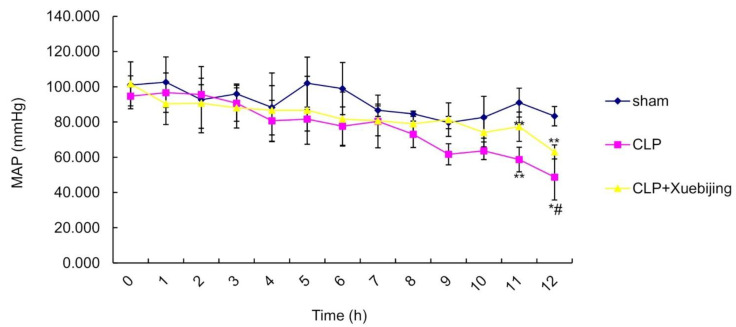
Xuebijing administration restored the mean arterial pressure (MAP) in cecal ligation and puncture (CLP)-induced septic rats. Rats were randomly divided into sham, CLP, and CLP + Xuebijing groups (n = 25/group). Each group was further divided into 2, 4, 6, 8, and 12 h subgroups (n = 5/subgroup). The MAP values were recorded in 3 randomly selected rats from the 12 h subgroup. Data are expressed as the mean ± standard deviation (SD). * *p* < 0.05, ** *p* < 0.01 vs. sham group; ^#^ *p* < 0.05 vs. CLP group; n = 3. MAP, mean arterial pressure; CLP, cecal ligation and puncture.

**Figure 2 jcm-11-06696-f002:**
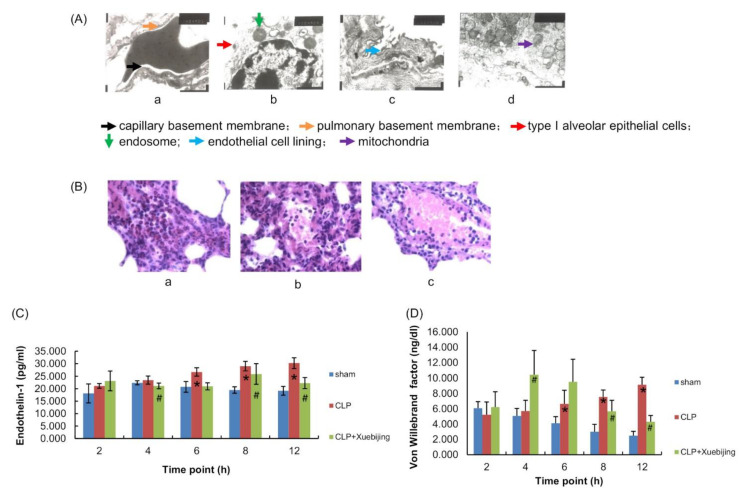
Xuebijing administration alleviated pulmonary endothelial injury in CLP rats. The lung tissue samples were collected from the rats in the 12 h subgroups. (**A**) Transmission microscopy was used to examine the structure of alveolar–capillary barrier. (**B**) Hematoxylin and eosin staining were conducted to examine pulmonary inflammation. (**C**,**D**) Blood samples were collected at 2, 4, 6, 8, and 12 h after surgery from each subgroup. The plasma levels of endothelin-1 and von Willebrand factor were measured. Data are expressed as the mean ± SD. * *p* < 0.05 vs. sham group; ^#^ *p* < 0.05 vs. CLP group; n = 5. CLP, cecal ligation and puncture.

**Figure 3 jcm-11-06696-f003:**
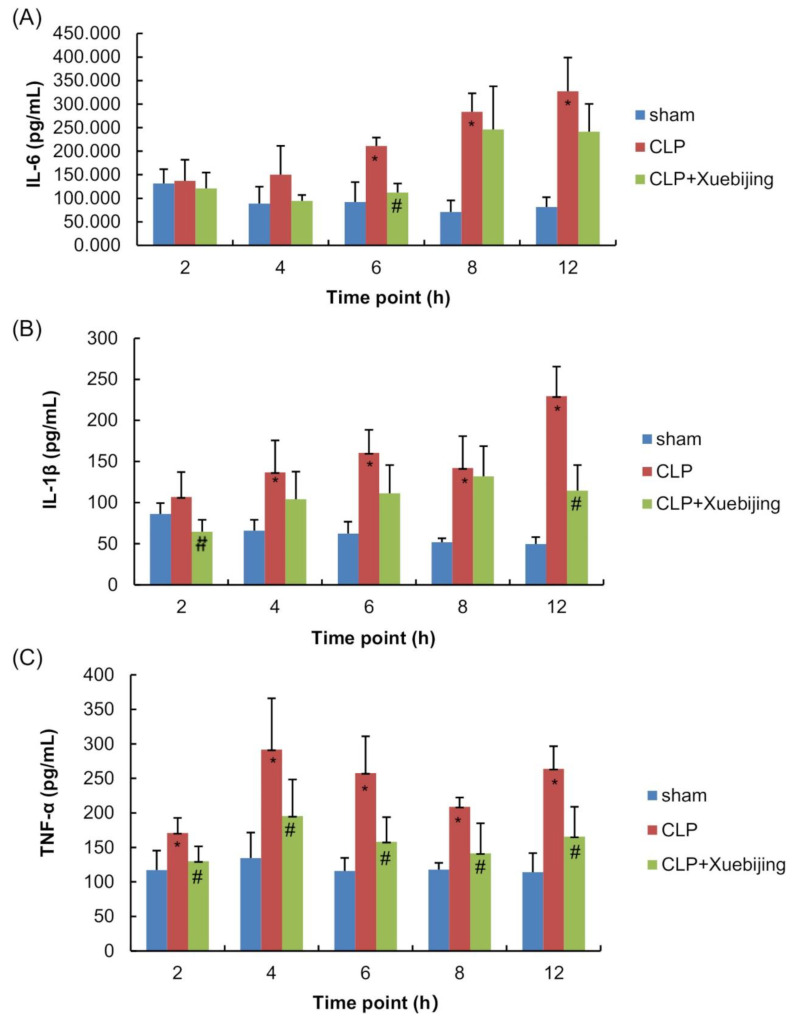
Xuebijing administration reduced circulating proinflammatory cytokine levels in CLP rats. Blood samples were collected at 2, 4, 6, 8, and 12 h after surgery from each subgroup. ELISA was performed to measure the plasma levels of interleukin 6 (**A**), interleukin 1β (**B**), and TNF-α (**C**) in rats. Data are expressed as the mean ± SD. * *p* < 0.05 vs. sham group, ^#^ *p* < 0.05 vs. CLP group; n = 5. CLP, cecal ligation and puncture; TNF-α, tumor necrosis factor α.

**Figure 4 jcm-11-06696-f004:**
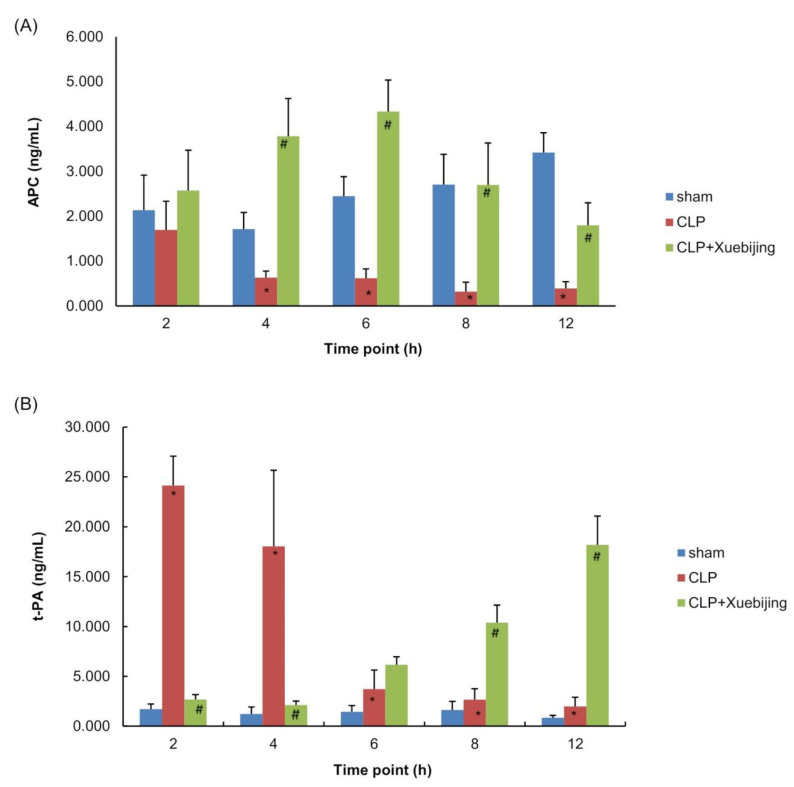
Xuebijing administration reversed CLP-induced alterations in circulating active protein C (APC) and tissue-type plasminogen activator (t-PA) levels in rats. Blood samples were collected at 2, 4, 6, 8, and 12 h after surgery from each subgroup. ELISA was performed to measure the plasma levels of APC (**A**) and t-PA (**B**) in rats. Data are expressed as the mean ± SD. * *p* < 0.05 vs. sham group, ^#^ *p* < 0.05 vs. CLP group; n = 5. CLP, cecal ligation and puncture; APC, active protein C; t-PA, tissue-type plasminogen activator.

## Data Availability

The authors confirm that the data supporting the findings of this study are available within the article and its supplementary materials.

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
