# Peer review of "Xuebijing Administration Alleviates Pulmonary Endothelial Inflammation and Coagulation Dysregulation in the Early Phase of Sepsis in Rats"

_jcm, 2022, doi:10.3390/jcm11226696_

Round 1

Reviewer 1 Report

CRITIQUE

The submitted manuscript for review is titled, “Xuebijing administration alleviates pulmonary endothelial inflammation and coagulation dysregulation in the early phase of sepsis in rats.”  Xuebijing (XBJ) is a traditional Chinese medicine preparation consisting of five Chinese herbs. Several types of biochemical compounds have been found in XBJ, including amino acids, phenolic acids, flavonoid glycoside, terpene glycoside, and phthalide [Phytochem Anal. 2011;22:330–338].

XBJ in sepsis has already been extensively studied experimentally and clinically. In fact, in 2004, XBJ injection was approved by the Chinese National Medical Products Administration (formerly the China Food and Drug Administration) specifically for the treatment of sepsis and associated organ dysfunction [Acta Pharm Sin B. 2019;9(5):1035–1049]. Furthermore, Chinese best practice guidelines and expert consensus opinion identifies XBJ injection administered together with an appropriate broad-spectrum antibiotic or antibiotic combination as the preferred treatment regimen for sepsis.  However, specific mechanisms for XBJ therapeutic effects have not been completely defined.

Of note, during the COVID-19 pandemic, interest in XBJ expanded further, when reports began to appear suggesting that XBJ injections may improve outcomes for this devastating SARS-CoV-2 infection. XBJ was found to shorten hospital and ICU LOS and reduce 28-day mortality rate [Aging Dis. 2021;12(8):1850-1856; Pharmacol Ther. 2021;225:107843; [Front Pharmacol. 2018;9:743; J Ethnopharmacol. 2018;224:512–521; Am J Emerg Med. 2017;35:285–291; Chin J Emerg Med. 2013;22:130–135; Chin Crit Care Med. 2015;27:465–470].]. Also, the efficacy of XBJ in patients with severe community-acquired pneumonia was demonstrated by a multicenter randomized placebo control trial published in 2019 [Crit. Care Med. 2019;47(9):e735–e743].

Numerous clinical and laboratory studies have suggested that XJB attenuates a systemic autoinflammatory response and maintains physiological functions of vital organs by inhibiting release of proinflammatory cytokines, modulating dysregulated coagulation and fibrinolytic cascades, inhibiting complement activation, and preventing systemic injury of the vascular endothelium [J Surg Res. 2016;202:147–154; J Ethnopharmacol. 2013;147:426–433; Chin J TCM WM Crit Care. 2016;23:554–557; Evid Based Complement Alternat Med. 2015;2015:860259; Front Pharmacol. 2018;9:300; J Biochem. 2018;164:427–435].

The authors posit XJB attenuates sepsis-induced injury to pulmonary endothelium, which the authors examine utilizing a rat cecal ligation and puncture (CPL) model of septicemia.  The authors’ experimental strategy is focused, carefully organized and well-controlled.  The strength of the study is in the histology presented; the biochemical data are less helpful, are not novel, and contribute little.  Nonetheless, the manuscript is exceptionally well-written and of appropriate length.  None of the figures are superfluous; the Discussion is concise.

However, the authors’ analysis of pulmonary endothelial cell dysfunction is arguably insufficient, and, therefore, this study at this time must be considered incomplete.

SPECIFIC COMMENTS

Hypothesis:

The endothelial glycocalyx is a heparan sulfate-rich layer of glycosaminoglycans and associated proteoglycans that lines the micro- and macrovascular intima. Sepsis-associated degradation of the pulmonary endothelial glycocalyx has been shown to contribute to septic lung injury [Am J Respir Cell Mol Biol. 2017;56(6):727–737]. For example, sepsis induces TNF-alpha-mediated endothelial heparanase activation degrades heparan sulfates in the glycocalyx leading directly to acute lung injury, which is characterized by barrier disruption and pulmonary edema [Crit Care. 2019;23:259].

The authors should examine glycocalyx degradation after CLP in animals treated with XBJ or vehicle.  Suggest ELISA on retained plasma samples (Rat Syndecan 1 (SDC1) ELISA Kit, catalog number: MBS2703971; https://www.mybiosource.com/rat-elisa-kits/syndecan-1-sdc1/2703971; MyBioSource, San Diego, CA). The authors might also consider immunohistochemistry with rat antisyndecan-1 monoclonal antibody (BD, PharmingenTM) then 1:500 Alexa Fluor goat anti-rat IgG (H+L, Invitrogen) on paraffin embedded tissue if these still remain.

Page 2, lines 72-76:

It is unclear why pulmonary epithelial cells are included here.  Was endothelial cell meant

Page 2, line 92:

Presumably the cecum was ligated prior to puncture. Please confirm.

Page 3, line 96:

When did the hour infusion of XBJ start relative to the CPL?

Page 8, lines 276-279:

Omit reporting on mitochondria number, considered by many to always be a meaningless observation.

Author Response

Dear reviewer :

We are very grateful to your encouraging and thoughtful comments and suggestions regarding our original submission. In response to these comments, we have made a number of modifications to our manuscript. Below we detail these specific modifications with the specific comments (in blue text) followed by our response (in black). We hope you will find this manuscript improved following these changes and more suitable for publication in J. Clin. Med.

SPECIFIC COMMENTS

Hypothesis:

The endothelial glycocalyx is a heparan sulfate-rich layer of glycosaminoglycans and associated proteoglycans that lines the micro- and macrovascular intima. Sepsis-associated degradation of the pulmonary endothelial glycocalyx has been shown to contribute to septic lung injury [Am J Respir Cell Mol Biol. 2017;56(6):727–737]. For example, sepsis induces TNF-alpha-mediated endothelial heparanase activation degrades heparan sulfates in the glycocalyx leading directly to acute lung injury, which is characterized by barrier disruption and pulmonary edema [Crit Care. 2019;23:259].

  1. i) The authors should examine glycocalyx degradation after CLP in animals treated with XBJ or vehicle.  Suggest ELISA on retained plasma samples (Rat Syndecan 1 (SDC1) ELISA Kit, catalog number: MBS2703971; https://www.mybiosource.com/rat-elisa-kits/syndecan-1-sdc1/2703971; MyBioSource, San Diego, CA). The authors might also consider immunohistochemistry with rat antisyndecan-1 monoclonal antibody (BD, PharmingenTM) then 1:500 Alexa Fluor goat anti-rat IgG (H+L, Invitrogen) on paraffin embedded tissue if these still remain.

Responses: Thank you very much for your kind suggestion. Due to the small blood volume of rats, the blood samples collected in this study were used to measure more cytokines, and there were no remaining plasma samples available. In the process of research, the tissue samples were processed with the help of professors from the pathology department and the electron microscope room of our hospital. Only the final pictures were provided to me and the tissue samples were destroyed after the research was completed. Unfortunately, more observations could not be made. However, according to your suggestion, I will conduct more in-depth research on the inflammatory reaction of other organs and the mechanism of endothelial damage in Xuebijing, and improve the research on blood and tissue samples related to the degradation of pulmonary endothelial glycocalyx in combination with your suggestion.

  1. ii) Page 2, lines 72-76:

It is unclear why pulmonary epithelial cells are included here. Was endothelial cell meant

Responses: Thank you very much for your kind suggestion. We have modified it in the revised manuscript. 

iii) Page 2, line 92:

Presumably the cecum was ligated prior to puncture. Please confirm.

 Responses: Thank you very much for your kind suggestion. We have modified it in the revised manuscript.

  1. iv) Page 3, line 96:

When did the hour infusion of XBJ start relative to the CPL?

Responses: Thank you very much for your kind question. Xuebijing injection was administered after closing the abdominal cavity. We have modified it in the revised manuscript.

  1. v) Page 8, lines 276-279:

Omit reporting on mitochondria number, considered by many to always be a meaningless observation.

Responses: Thank you very much for your kind suggestion. We have modified it in the revised manuscript.

Reviewer 2 Report

Content suggestions:

1.               Why did the authors not use healthy controls ?

2.               Why did they not to test further markers of coagulopathy in sepsis – antithrombin level, D-dimer levels, thrombin-antithrombin complex...?

3.               How can the authors explain the decrease in APC after the administration of Xuebijing ? Usually, derangement of protein C system and decreased PC is vulnerable to depletion with acute ischemic hepatitis... Moreover, its levels are supposed to be markers of mortality.

Author Response

Content suggestions:

i) Why did the authors not use healthy controls ?

Responses: Thank you very much for your kind suggestion. This study simulated the scene of ICU treating severe sepsis patients to the maximum extent, and selected the sham operation group of abdominal opening and closing as the control, instead of using healthy rats as the control, in order to reduce the deviation and interference caused by surgical operation and sedation on the body.

ii) Why did they not to test further markers of coagulopathy in sepsis – antithrombin level, D-dimer levels, thrombin-antithrombin complex...?

Responses: Thank you very much for your kind suggestion. The D-dimer is a commonly used clinical indicator to evaluate the transient hypercoagulability in the early stage of sepsis and fibrinolysis hyperactivity after microthrombosis. Considering that the abnormality of the D-dimer may not be obvious in the early stage of inflammation after CLP surgery in rats, TPA measurement was selected to reflect the start of fibrinolysis. Because a limited number of plasma samples are used for the examination of other related cytokines, it is regrettable that no complete determination of coagulation related indicators has been carried out.

iii)How can the authors explain the decrease in APC after the administration of Xuebijing ? Usually, derangement of protein C system and decreased PC is vulnerable to depletion with acute ischemic hepatitis... Moreover, its levels are supposed to be markers of mortality.

Responses: Thank you very much for your kind question. In the early stage of sepsis treatment, some doctors tried to use APC to treat the hypercoagulable and microthrombotic state in the early stage of sepsis, but the effect was not good, even leading to severe bleeding symptoms. This also confirmed that the period of hypercoagulability and microthrombosis in the pathophysiological process of sepsis is short. In our study, through the pathophysiological reaction of sepsis caused by surgery, we can observe the transient process of hypercoagulability and microthrombosis in the early stage of inflammatory reaction. It can be seen that in the control group, the activation of APC is inhibited within 8 hours after surgery, and it starts to increase slightly at 12 hours, indicating that the mechanism of the PC system against early hypercoagulability and microthrombosis is inhibited. Xuebijing injection, as a compound preparation of traditional Chinese medicine, helps to increase APC in the early stage to fight against early hypercoagulable state, while in the later stage, when the body is about to enter a period of hyperfibrinolysis, it helps to reduce the level of APC in a timely manner to prevent the risk of body bleeding from further increasing due to excessive activation of PC system during hyperfibrinolysis in the late stage of sepsis, which also reflects the dual regulatory role of traditional Chinese medicine in "activating blood" and "stopping blood". However, the mechanism by which traditional Chinese medicine can play a two-way regulatory role really needs further research

Round 2

Reviewer 1 Report

Manuscript satisfactorily modified